

# Interaction effects of significant risk factors on low bone mineral density in ankylosing spondylitis

Wenting Sun[1,*], Wenjun Mu[2,*], Caroline Jefferies[3], Thomas Learch[4], Mariko Ishimori[3], Juan Wu[2], Zeran Yan[5,6], Nan Zhang[5,6], Qingwen Tao[5,6], Weiping Kong[5,6], Xiaoping Yan[5,6] and Michael H. Weisman[7]

[1] China Academy of Chinese Medical Sciences, Beijing, China
[2] Beijing University of Chinese Medicine, Beijing, China
[3] Division of Rheumatology, Department of Medicine, Cedars-Sinai Medical Center, Los Angeles, United States of America
[4] Department of Radiology, Cedars-Sinai Medical Center, Los Angeles, United States of America
[5] Beijing Key Lab for Immune-Mediated Inflammatory Diseases, China-Japan Friendship Hospital, Beijing, China
[6] Department of TCM Rheumatology, China-Japan Friendship Hospital, Beijing, China
[7] Stanford University, Stanford, California, United States of America
* These authors contributed equally to this work.

Corresponding author
Weiping Kong, kwptgzy@163.com

## ABSTRACT

**Background:** To analyze individually and interactively critical risk factors, which are closely related to low bone mineral density (BMD) in patient with ankylosing spondylitis (AS).

**Methods:** A total of 249 AS patients who visited China-Japan Friendship Hospital were included in this training set. Patients with questionnaire data, blood samples, X-rays, and BMD were collected. Logistic regression analysis was employed to identify key risk factors for low BMD in different sites, and predictive accuracy was improved by incorporating the selected significant risk factors into the baseline model, which was then validated using a validation set. The interaction between risk factors was analyzed, and predictive nomograms for low BMD in different sites were established.

**Results:** There were 113 patients with normal BMD, and 136 patients with low BMD. AS patients with hip involvement are more likely to have low BMD in the total hip, whereas those without hip involvement are more prone to low BMD in the lumbar spine. Chest expansion, mSASSS, radiographic average grade of the sacroiliac joint, and hip involvement were significantly associated with low BMD of the femoral neck and total hip. Syndesmophytes, hip involvement and higher radiographic average grade of the sacroiliac joint increases the risk of low BMD of the femoral neck and total hip in an additive manner. Finally, a prediction model was constructed to predict the risk of low BMD in total hip and femoral neck.

**Conclusions:** This study identified hip involvement was strongly associated with low BMD of the total hip in AS patients. Furthermore, the risk of low BMD of the femoral neck and total hip was found to increase in an additive manner with the presence of syndesmophytes, hip involvement, and severe sacroiliitis. This finding may help rheumatologists to identify AS patients who are at a high risk of developing low BMD and prompt early intervention to prevent fractures.

## INTRODUCTION

Ankylosing spondylitis (AS) is a chronic inflammatory disease that primarily affects the axial skeleton (*Mauro et al., 2021*; *Klavdianou, Tsiami & Baraliakos, 2021*). The main feature of the disease is new bone formation, including syndesmosis, syndesmophytes formation, fusion of the sacroiliac joints, and ankylosis of the spine (*Hwang, Ridley & Reveille, 2021*). Low bone mineral density (BMD) is considered to be one of the most common complications in AS, which occurs in a range of 19% to 62% of patients undergoing screening (*van der Weijden et al., 2012*; *Hinze & Louie, 2016*; *Klingberg et al., 2012*; *Ramírez et al., 2018*). However, due to the relatively subtle symptoms of low BMD for AS patients, early detection is not easily achievable. Many physicians are unaware of the increased risk of low BMD in AS, which undoubtedly contributes to delayed diagnosis and increased fracture risk.

The risk factors and pathophysiology mechanism on low BMD in AS patient are unclear. Previous studies have shown that inflammation, disease course, disease activity, the release of inflammatory cytokines, mechanical factors, radiological damage may be related to low BMD in AS patients (*Malochet et al., 2017*; *Kim et al., 2022*; *Bautista-Aguilar et al., 2021*). To date, the risk factors of low BMD at different sites remain controversial. For example, *Kaya et al. (2009)* reported that as the disease progressed, BMD decreased in the femoral neck but increased in the lumbar spine. However, *Wu et al. (2021)* reported that lumbar BMD was not related to the course of AS. The most compelling reason for the apparent observed inconsistencies may be the complex course, over time, of low BMD in AS, which is unlikely to be solely influenced by one single predictor. Currently, there is no adequate explanation for the common risk factors for low BMD at different sites or for established interactive relationships among these risk factors.

The objectives of this study were to analyze individually and interactively critical risk factors, which are closely related to low BMD in patient with AS, and establish a nomogram prediction model to guide AS patients in preventing the occurrence of low BMD.

## MATERIALS AND METHODS

### Research subjects

A total of 249 patients from July 2012 to November 2018 recruited at the China-Japan Friendship Hospital were used as the training set. Another independently cohort including 140 patients from March 2019 to August 2022 were used as the validation set. This study employs continuous enrollment to mitigate selection bias. In this research, all included patients met the modified New York criteria for AS (*Van der Linden, Valkenburg & Cats, 1984*). Exclusion criteria: (1) the patient had undergone hip replacement surgery; (2) Suffering from other autoimmune diseases, including but not limited to inflammatory

bowel disease, psoriasis, ichthyosis, *etc*; (3) The patient has serious basic diseases, such as severe malnutrition, liver and kidney failure, *etc*; (4) The patient did not agree to participate in this study. All patients included in this study were evaluated by doctors to see if they met the inclusion criteria and signed the informed consent. The study has been reviewed and approved by ethics committees, including the research ethics committee of China-Japan Friendship Hospital (approval No. 2017-67) and the ethics committee of cedar Sinai Medical Center (approval No. pro00048849), and was conducted in accordance with the declaration of Helsinki.

## Data collection/measurements

The questionnaires were administered by investigators with experience in epidemiological research. Patients' individual data, including gender, age, body mass index (BMI), current medication status, smoking, alcohol-related conditions, family history, onset age, and sports activities, were collected. The functional status, disease activity and severity in patients with AS were obtained by filling in the Bath Ankylosing Spondylitis Disease Activity Index (BASDAI) and Bath Ankylosing Spondylitis Functional Index (BASFI) questionnaires (*Zochling, 2011*).

All patients were evaluated by the same rheumatologist. The evaluation scope included physical examination, modified-Schober score, chest expansion score, and the Bath Ankylosing Spondylitis Metrology Index (BASMI) (*Calin et al., 1994*; *Song et al., 2009*; *Jenkinson et al., 1994*). The visual analog scale (VAS, 0–10cm) was used to assess the night pain and the patient global assessment (PGA) (*Sieper et al., 2009*).

The New York classification criteria was used to grade the degree of sacroiliac joint, and the classification criteria were normal (0) to most serious (4). The diagnosis of AS was unilateral grade 3, unilateral grade 4 or bilateral grade 2 (*Van der Linden, Valkenburg & Cats, 1984*). A lateral radiograph of the cervical, thoracic and lumbar spine were collected, and the modified ankylosing spondylitis score (mSASSS) was used to evaluate the AS-related changes. The scoring standard of mSASSS are as follow the previously published study (*van der Heijde et al., 2019*). The mSASSS was scored by a musculoskeletal radiologist and a cross-trained rheumatologist. Cohen's kappa coefficient was used to analyze the consistency of the two doctors' scores on the study subjects. When Cohen's kappa ($\kappa$) coefficients were >0.85, it indicates that the consistency between researchers is good.

Hip involvement was evaluated by an experienced rheumatologist, including restricted range of motion, pain and Bath Ankylosing Spondylitis Radiology Hip Index (BASRI-hip). The BASRI-hip scoring method was refer to previous published study (*Konsta et al., 2023*). Radiographic hip joint involvement was defined by at least 1 score in the BASRI-hip scoring system (*MacKay et al., 2000*).

The disease activity score (ASDAS) for ankylosis was calculated using the formula in order to better assess the patient's disease activity (*Deodhar et al., 2022*). The calculation of ASDAS-CRP refers to previous literature (*Ørnbjerg et al., 2022*). To evaluate the disease activity, the disease activity score (ASDAS) of ankylosis was calculated using the

ASDAS-CRP formula, and the use of formula refers to published literature (*Zochling, 2011*).

At the time of patient enrollment, blood samples from AS patients were collected and analyzed using standard laboratory techniques, and BMD was measured. Before serum samples were collected, patients fasted overnight (at least 8 h). Indicators reflecting inflammation, including ESR and CRP, were collected. Other laboratory indicators closely related to AS, such as HLA-B27, were also recorded. BMD (g/cm2) was measured using dual-energy X-ray absorptiometry (DXA). The detection range of BMD includes: lumbar spine (L1-L4), femoral neck, and total hip. In this study, patients with a measured bone density T-scores < −1 at either site were defined as low BMD (*Cabrera et al., 2018*). In addition, the low BMD group was also divided into osteoporosis and osteopenia. Osteopenia and osteoporosis are defined according to the World Health Organization standards (*Kanis, 1994*).

## Statistical analysis

Continuous variables are represented using the median, range, and/or mean, along with the standard deviation (SD) where appropriate. Categorical variables are presented as frequencies or percentages. To ensure data integrity, we endeavor to employ mean or median imputation whenever possible to address potential data missing issues. In cases of relatively symmetrical data distributions without conspicuous outliers, we typically opt for mean imputation. Conversely, when data distributions exhibit pronounced skewness or contain outliers, we are inclined to utilize median imputation. To assess differences between groups, appropriate statistical tests such as the chi-square test, t-test, and rank sum test were employed. The correlation coefficient was calculated using Spearman analysis. Logistic regression analysis was performed to identify significant risk factors associated with low BMD, both before and after adjusting for confounding factors. The effect sizes are reported as odds ratios (ORs) with corresponding 95% confidence intervals (CIs). We select variables based on the results of logistic regression and clinical relevance, ensuring that the inclusion of each feature is a reasonable explanation, and establish predictive model based on these variables. A nomogram for low BMD across different sites was developed, and its accuracy was evaluated using the concordance index. Akaike information criterion (AIC), Bayesian information criterion (BIC), and likelihood ratio (LR) test are used for model calibration, while net reclassification improvement (NRI), integrated discrimination improvement (IDI), and area under the receiver operating characteristic (AUROC) are employed for model discrimination. Clinical utility is assessed using decision curve analysis (DCA), and the model is further validated using validation set data. All statistical analyses were carried out using STATA software, special edition (version 14.0, STATA Corp, TX). The nomogram was constructed using the R language (version 3.5.2). By calculating the variance inflation factor (VIF), we determine whether there is collinearity among the factors. A VIF below 10 indicates no significant collinearity and can be used to create a nomogram (*Cheng et al., 2022*). The nomogram associates each variable with its corresponding score, and the sum of scores for all variables is defined as the total score. By drawing a vertical line from the axis of the total score, the

**Table 1 Baseline characteristics of the training set.**

| Variables | All (N = 249) | Normal BMD (N = 132) | Low BMD (N = 117) | P value |
|---|---|---|---|---|
| **Demographic variables** | | | | |
| Male | 194 (77.9%) | 98 (74.2%) | 96 (82.1%) | 0.138 |
| Age, years | 33.7 (10.5) | 34.3 (10.4) | 33.1 (10.6) | 0.348 |
| BMI, kg/m$^2$ | 23.1 (3.2) | 23.4 (3.0) | 22.7 (3.4) | 0.044 |
| Sport | 38 (15.4%) | 20 (15.2%) | 18 (15.4%) | 0.959 |
| Family history | 55 (21.1%) | 31 (23.5%) | 24 (20.5%) | 0.573 |
| Diagnosis duration, years | 6.3 (5.0) | 5.8 (4.8) | 6.8 (5.2) | 0.164 |
| Symptoms duration, years | 10.0 (6.7) | 9.7 (6.9) | 10.2 (6.5) | 0.291 |
| Onset age, years | 23.8 (9.9) | 24.6 (9.7) | 22.9 (10.1) | 0.106 |
| Smoking index | 48.6 (139.7) | 46.8 (149.1) | 50.6 (128.6) | 0.462 |
| Smoke duration, years | 3.1 (6.8) | 2.8 (6.14) | 3.5 (7.5) | 0.468 |
| Cigarettes per day | 3.5 (7.3) | 3.3 (7.7) | 3.6 (6.8) | 0.389 |
| Ever smoking | 65 (26.1%) | 31 (23.5%) | 34 (29.1%) | 0.317 |
| Current smoking | 62 (24.9%) | 29 (22.0%) | 33 (28.2%) | 0.256 |
| Alcohol duration | 3.0 (6.8) | 3.1 (6.8) | 2.9 (6.9) | 0.701 |
| Alcohol history | 50 (20.1%) | 26 (19.7%) | 24 (20.5%) | 0.873 |
| Daily alcohol | 55 (22.1%) | 29 (22.0%) | 26 (22.2%) | 0.962 |
| Alcohol frequency | 0.3 (0.8) | 0.3 (0.7) | 0.3 (0.8) | 0.939 |
| **Current medication status** | | | | |
| Patients on TNF inhibitor | 8 (3.2%) | 3 (2.3%) | 5 (4.3%) | 0.372 |
| Patients on NSAIDs | 137 (55.0%) | 71 (53.8%) | 66 (56.4%) | 0.678 |
| Patients on cDMARDs | 54 (21.7%) | 23 (17.4%) | 31 (26.5%) | 0.083 |

Note:
BMD, bone mineral density; BMI, body boss index; TNF, tumour necrosis factor; cDMARDs, conventional DMARDs.

estimated probability of occurrence can be obtained. Statistical significance was considered at a threshold of $P < 0.05$.

# RESULTS

## Patient characteristics

A total of 249 patients from the training set participated in this study, including 194 males and 55 females. The average age was 34 ± 11 years, the onset age was 24 ± 10 years, and the diagnostic duration was 6 ± 5 years. Among them, 132 patients (53.0%) had normal BMD, and 117 patients (47.0%) had low BMD. BMI was significantly different between the normal BMD group and the low BMD group ($P < 0.05$), while other baseline characteristics were similar (all $P > 0.05$). The characteristics of study patients in the training set are presented in Table 1.

The disease-related variables and BMD of the study patients are summarized in Table 2. BASMI, chest expansion, radiographic average grade of the sacroiliac joint, hip involvement, and ASDAS-CRP showed a significant difference between the normal BMD

**Table 2 Disease-related variables and bone mineral density of the training set.**

| Variables | All (N = 249) | Normal BMD (N = 132) | Low BMD (N = 117) | P value |
|---|---|---|---|---|
| **Disease-related variables** | | | | |
| BASDAI, score | 3.1 (1.3) | 3.0 (1.2) | 3.3 (1.4) | 0.134 |
| BASFI, score | 1.4 (1.6) | 1.4 (1.5) | 1.5 (1.7) | 0.565 |
| BASMI, score | 1.9 (2.0) | 1.7 (2.0) | 2.2 (2.0) | 0.027 |
| Night pain, score | 4.0 (2.0) | 3.8 (1.9) | 4.2 (2.0) | 0.104 |
| PGA, score | 3.9 (2.0) | 3.8 (2.0) | 4.1 (1.9) | 0.275 |
| Chest expansion, cm | 4.4 (2.0) | 4.7 (1.8) | 4.1 (2.1) | 0.030 |
| modified-Schober, score | 5.0 (1.9) | 5.2 (1.8) | 4.8 (1.9) | 0.083 |
| HLA-B27 Positive | 223 (89.6%) | 117 (88.6%) | 106 (90.6%) | 0.613 |
| Sacroiliitis average, score | 3.0 (0.8) | 2.8 (0.8) | 3.2 (0.8) | <0.001 |
| mSASSS, score | 13.2 (20.6) | 11.1 (19.7) | 14.9 (21.2) | 0.142 |
| Hip involvement | 78 (38.4%) | 35 (31.3%) | 43 (47.3%) | 0.020 |
| ESR, mm/h | 21.0 (19.7) | 19.0 (18.9) | 23.3 (19.6) | 0.051 |
| CRP, mg/L | 1.6 (1.9) | 1.6 (2.0) | 1.7 (1.9) | 0.142 |
| ASDAS-CRP, scores | 1.7 (0.7) | 1.6 (0.7) | 1.8 (0.8) | 0.032 |
| **BMD** | | | | |
| Lumbar spine | 1.1 (0.19) | 1.2 (0.17) | 1.0 (0.16) | <0.001 |
| Femoral neck | 0.9 (0.15) | 1.0 (0.11) | 0.8 (0.14) | <0.001 |
| Total hip | 0.9 (0.17) | 1.0 (0.14) | 0.8 (0.14) | <0.001 |

**Note:**

BMD, bone mineral density; BASDAI, Bath Ankylosing Spondylitis Disease Activity Index; BASMI, Bath Ankylosing Spondylitis Metrology Index; BASFI, Bath Ankylosing Spondylitis Functional Index; PGA, patient global assessment; HLA-B27, human leucocyte antigen B27; mSASSS, modified ankylosing spondylitis score; Sacroiliitis average, means average radiological grade of the sacroiliac joint.

and the low BMD groups (all $P < 0.05$). The baseline characteristics and disease-related variables in validation set were shown in Table S1.

## Prevalence of low BMD in different sites

Among all patients in the training set with low BMD, included 105 patients (89.7%) had osteopenia and 12 patients (10.3%) had osteoporosis. The prevalence of low BMD in different sites is shown in Table S2. The lumbar spine was the most common site for low BMD (29.3%), followed by the femoral neck (26.5%) and total hip (24.9%). For patients with hip involvement, the total hip was the most common site for low BMD (34.5%). For patients without hip involvement, the lumbar spine was the most common site for low BMD (16.7%).

## The association of BMD and mSASSS in AS patients with syndesmophytes

For AS patients in training set with syndesmophytes, increase in mSASSS was significantly associated with higher anteroposterior lumbar spine BMD ($r_s = 0.201$, $P = 0.024$) but not with femoral neck or total hip BMD ($r_s = -0.156$, $P = 0.081$; $r_s = -0.146$, $P = 0.102$, respectively) (Table S3).

### Identification of risk factors for low BMD in the femoral neck and total hip

We further studied the effect-size estimates of multiple examined factors in association with the risk of low BMD before and after adjusting for confounding factors for femoral neck and total hip BMD (Table 3). Based on univariate logistic regression analysis several factors associated with the development of low BMD in the femoral neck and total hip were found at a significance level of 5% (Table S4). After adjusting for age and gender, statistical significance was still existed in all factors. After multivariate adjustment, chest expansion, BASMI, mSASSS, BMI, average radiographic grades at the sacroiliac joint and hip involvement were recognized as risk factors for low BMD of the femoral neck ($P < 0.05$). Chest expansion, BASFI, mSASSS, ASDAS-CRP, diagnosis duration, BMI, night pain, average radiographic grades at the sacroiliac joint, PGA, and hip involvement were recognized as risk factors for low BMD of the total hip ($P < 0.05$).

### Prediction accuracy assessment

Basic and full models were constructed to evaluate the predictive performance of important factors associated with low BMD (Table 4). The full model included all the variables investigated, however, the basic model included all variables except for the significant risk factors identified by regression analyses. Calibration and discriminant statistics were applied to evaluate the prediction performance of the femoral neck and total hip significance factors which were added in the basic model. The prediction accuracy of the full model was significantly higher than that of the basic model. As shown by the comprehensive discriminant improvement, there were significant differences between the two models in predicting the performance of low BMD in the femoral neck and total hip ($P < 0.001$). For both the femoral neck and total hip, decision curve analysis suggested that our full model had superiority over the basic model for the fact that more clinical net benefits were obtained in a rather wide range of threshold probabilities when using full models than those when using the basic model (Fig. 1).

We applied the model to the validation set comprising 140 patients. The results demonstrated that the full model achieved an AUROC of 0.79 in the femoral neck prediction model and an AUROC of 0.80 in the total hip prediction model. The predictive performance of the validation set in Table S5.

### Interaction explorations

Since the occurrence of low BMD in AS patients is a complex process, the influence of any risk factor may be small when evaluated alone, but it may be more obvious when other risk factors are combined. To obtain more accurate information, combined with the outcomes of clinical and logistic regression analysis, we divided the variables that were relevant for the low BMD into groups and further explored the risk factors affecting low BMD in the femoral neck and total hip (Table 5).

Hip involvement, mSASSS, and the average radiographic grade of the sacroiliac joint were found to be significant risk factors associated with low BMD in the femoral neck and total hip. When the average radiological grade of the sacroiliac joint exceeds grade 3 or hip

**Table 3 Risk prediction for low BMD in AS patients.**

| Variables | Femoral neck | | | Total hip | | |
|---|---|---|---|---|---|---|
| | OR | 95% CI | *P* | OR | 95% CI | *P* |
| **Unadjusted** | | | | | | |
| Chest expansion | 0.81 | [0.70–0.94] | 0.005 | 0.86 | [0.74–0.99] | 0.038 |
| BASMI | 1.21 | [1.05–1.39] | 0.007 | 1.17 | [1.01–1.34] | 0.032 |
| BASFI | 1.04 | [0.88–1.24] | 0.640 | 1.15 | [0.98–1.36] | 0.095 |
| Total mSASSS | 1.02 | [1.01–1.03] | 0.003 | 1.02 | [1.00–1.03] | 0.011 |
| ASDAS-CRP | 1.22 | [0.84–1.78] | 0.286 | 1.45 | [0.99–2.12] | 0.057 |
| Diagnosis duration | 1.04 | [0.99–1.10] | 0.157 | 1.07 | [1.01–1.13] | 0.019 |
| BMI | 0.92 | [0.84–1.00] | 0.054 | 0.88 | [0.80–0.97] | 0.010 |
| Night pain | 0.96 | [0.83–1.11] | 0.598 | 1.16 | [1.00–1.34] | 0.048 |
| PGA | 1.03 | [0.90–1.19] | 0.662 | 1.16 | [1.01–1.35] | 0.041 |
| Sacroiliitis average | 2.08 | [1.44–3.02] | <0.001 | 1.90 | [1.31–2.75] | 0.001 |
| Hip involvement | 2.08 | [1.55–5.20] | 0.001 | 2.90 | [1.55–5.43] | 0.001 |
| **Age and gender adjusted** | | | | | | |
| Chest expansion | 0.82 | [0.71–0.95] | 0.008 | 0.85 | [0.73–0.99] | 0.031 |
| BASMI | 1.20 | [1.04–1.38] | 0.013 | 1.18 | [1.02–1.37] | 0.025 |
| BASFI | 1.03 | [0.87–1.22] | 0.729 | 1.15 | [0.97–1.36] | 0.099 |
| Total mSASSS | 1.02 | [1.00–1.03] | 0.016 | 1.02 | [1.01–1.04] | 0.009 |
| ASDAS-CRP | 1.25 | [0.86–1.82] | 0.245 | 1.47 | [1.00–2.16] | 0.051 |
| Diagnosis duration | 1.03 | [0.97–1.09] | 0.293 | 1.07 | [1.01–1.14] | 0.017 |
| BMI | 0.90 | [0.82–0.99] | 0.026 | 0.88 | [0.80–0.97] | 0.008 |
| Night pain | 0.97 | [0.84–1.12] | 0.692 | 1.16 | [1.01–1.35] | 0.041 |
| PGA | 1.05 | [0.91–1.21] | 0.547 | 1.17 | [1.01–1.35] | 0.036 |
| Sacroiliitis average | 2.20 | [1.49–3.24] | <0.001 | 2.03 | [1.38–2.99] | <0.001 |
| Hip involvement | 2.88 | [1.48–5.59] | 0.002 | 3.05 | [1.61–5.79] | 0.001 |
| **Multivariable adjusted** | | | | | | |
| Chest expansion | 0.79 | [0.67–0.93] | 0.005 | 0.81 | [0.69–0.95] | 0.011 |
| BASMI | 1.18 | [1.01–1.38] | 0.035 | 1.16 | [0.99–1.36] | 0.060 |
| BASFI | 1.08 | [0.89–1.31] | 0.424 | 1.22 | [1.01–1.47] | 0.044 |
| Total mSASSS | 1.02 | [1.00–1.04] | 0.016 | 1.02 | [1.00–1.04] | 0.014 |
| ASDAS-CRP | 1.38 | [0.91–2.09] | 0.133 | 1.62 | [1.06–2.48] | 0.027 |
| Diagnosis duration | 1.04 | [0.98–1.11] | 0.166 | 1.07 | [1.00–1.14] | 0.038 |
| BMI | 0.87 | [0.78–0.97] | 0.009 | 0.87 | [0.79–0.97] | 0.012 |
| Night pain | 1.01 | [0.86–1.18] | 0.946 | 1.21 | [1.03–1.42] | 0.018 |
| PGA | 1.09 | [0.93–1.27] | 0.278 | 1.22 | [1.04–1.43] | 0.016 |
| Sacroiliitis average | 2.09 | [1.38–3.17] | 0.001 | 2.01 | [1.32–3.05] | 0.001 |
| Hip involvement | 2.83 | [1.38–5.82] | 0.004 | 2.77 | [1.33–5.76] | 0.006 |

**Note:**
BASMI, Bath Ankylosing Spondylitis Metrology Index; BASFI, Bath Ankylosing Spondylitis Functional Index; BMI, body boss index; PGA, patient global assessment. OR, odds ratio; 95% CI, 95% confidence interval. Sacroiliitis average, means average radiological grade of the sacroiliac joint. *P* values were calculated before and after adjusting for age, gender, HLA-B27, smoking index, smoking duration, cigarettes per day, current smoking, alcohol history and alcohol duration. In multivariable adjusted model, risk prediction of each adjusted factor was calculated by adjusting for the other factors.

**Table 4 Prediction accuracy gained by adding the identified significant factors for low BMD in training set.**

| Statistic | Femoral neck | | Total hip | |
|---|---|---|---|---|
| | Basic model | Full model | Basic model | Full model |
| Calibration | | | | |
| AIC | 302 | 226 | 286 | 213 |
| BIC | 395 | 333 | 366 | 320 |
| LR test ($\chi^2$) | Ref. | 20.93 | Ref. | 25.11 |
| LR test (*P* value) | Ref. | 0.002 | Ref. | <0.001 |
| Discrimination | | | | |
| NRI (*P* value) | Ref. | 0.013 | Ref. | 0.022 |
| IDI (*P* value) | Ref. | <0.001 | Ref. | <0.001 |
| AUROC | 0.62 | 0.74 | 0.62 | 0.77 |
| AUROC (*P* value) | 0.003 | | 0.002 | |

**Note:**
AIC, Akaike information criterion; BIC, Bayesian information criterion; LR, likelihood ratio; NRI, net reclassification improvement; IDI, integrated discrimination improvement; AUROC, area under the receiver operating characteristic; Ref., reference.

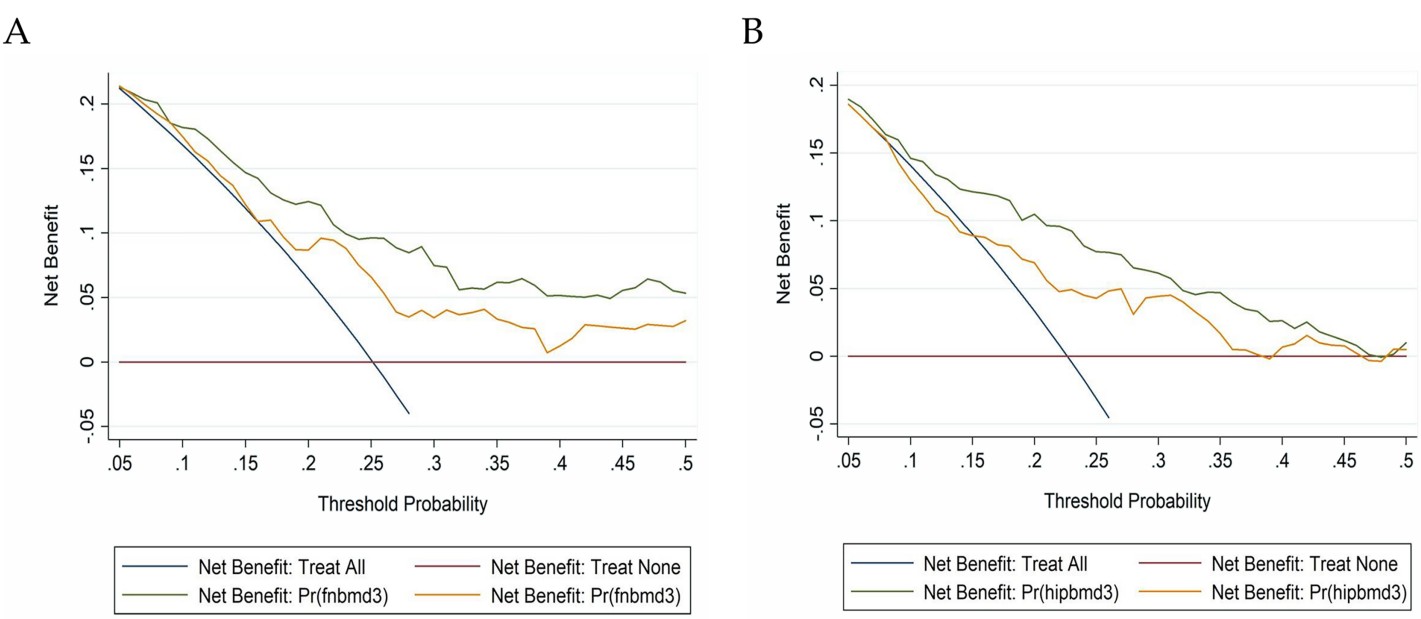

**Figure 1 Net benefits gained by the significant factors identified for low BMD in AS patients in decision curve analysis at two different sites.**
(A) Femoral neck; (B) total hip.

involvement, the presence of syndesmophytes (defined as at least one vertebral corner mSASSS score ≥ 2 (*van der Heijde et al., 2019*)) further increased the risk of low BMD in the femoral neck and total hip (both *P* < 0.05).

Notably, hip involvement not only interacted with the presence of syndesmophytes but also with the average radiological grade of the sacroiliac joint. When the average radiological grade of the sacroiliac joint exceeds grade 3, hip involvement increased the risk

**Table 5 The interaction of three significant factors identified for low BMD of the femoral neck and total hip in AS patients.**

| Interaction items | Femoral neck | | | Total hip | | |
|---|---|---|---|---|---|---|
| | OR | 95% CI | P | OR | 95% CI | P |
| No syndesmophytes/Sacroiliitis average ≤ 3 | Ref. | | | Ref. | | |
| No syndesmophytes/Sacroiliitis average > 3 | 1.40 | [0.51–3.81] | 0.513 | 3.68 | [1.34–10.12] | 0.012 |
| Syndesmophytes present/Sacroiliitis average ≤ 3 | 0.43 | [0.10–1.79] | 0.243 | 1.11 | [0.29–4.20] | 0.883 |
| Syndesmophytes present/Sacroiliitis average > 3 | 2.29 | [1.02–5.15] | 0.045 | 3.35 | [1.37–8.18] | 0.008 |
| No syndesmophytes/Hip involvement = 0 | Ref. | | | Ref. | | |
| No syndesmophytes/Hip involvement = 1 | 2.22 | [0.70–7.01] | 0.174 | 4.74 | [1.48–15.20] | 0.009 |
| Syndesmophytes present/Hip involvement = 0 | 1.14 | [0.40–3.20] | 0.809 | 1.92 | [0.64–5.77] | 0.244 |
| Syndesmophytes present/Hip involvement = 1 | 3.51 | [1.33–9.30] | 0.012 | 3.55 | [1.21–10.36] | 0.021 |
| Sacroiliitis average ≤ 3/Hip involvement = 0 | Ref. | | | Ref. | | |
| Sacroiliitis average ≤ 3/Hip involvement = 1 | 1.33 | [0.24–7.46] | 0.746 | 4.43 | [0.97–20.22] | 0.055 |
| Sacroiliitis average > 3/Hip involvement = 0 | 1.31 | [0.48–3.57] | 0.596 | 2.57 | [0.88–7.56] | 0.086 |
| Sacroiliitis average > 3/Hip involvement = 1 | 3.73 | [1.48–9.39] | 0.005 | 4.78 | [1.71–13.37] | 0.003 |

**Note:**
OR, odds ratio; 95% CI, 95% confidence interval; Ref., reference; Sacroiliitis average, means average radiological grade of the sacroiliac joint; Syndesmophytes present, defined as at least one vertebral corner mSASSS score ≥2.

of low BMD in the femoral neck and total hip (both $P < 0.05$). This finding implies that for AS patients with hip involvement, the combination of severe sacroiliitis or syndesmophytes significantly increase the risk of low BMD at the femoral neck and total hip.

### Prediction model

Finally, a nomogram was constructed to predict the risk of low BMD in AS patients based on significant factors that were identified in the femoral neck and total hip (Fig. 2). Nomogram's important factors were analyzed by positive logistic regression at a significance level of 5%. For a specific patient, each indicator has specific values, mapped onto the "Points" scale to obtain individual scores per indicator. Summing all scores gives a total. Locate this total on the "Total Points" and map to the "Risk" scale to determine patient low BMD risk. For example, assuming a female (20 points) AS patient with an onset age of 20 (10 points), BMI of 18 (82 points), BASFI of 5 (64 points), chest expansion of 3 (30 points), sacroiliitis average of four scores (10 points), mSASSS scores of 18 (10 points) and hip involvement (50 points), the possibility of low BMD of total hip was estimated to be 70%.

### DISCUSSION

Low BMD is the most common comorbidity of AS due to multiple factors that disrupt bone metabolic balance. It increased fracture risk in AS patients, therefore, identifying risk factors is of great importance for the prevention of low BMD. The main purpose of this study is to investigate individually and interactively critical risk factors for low BMD in AS patients at different sites and to establish predictive nomogram models reflecting the data from our subjects. To our knowledge, this is the first study to predict the risk factors for
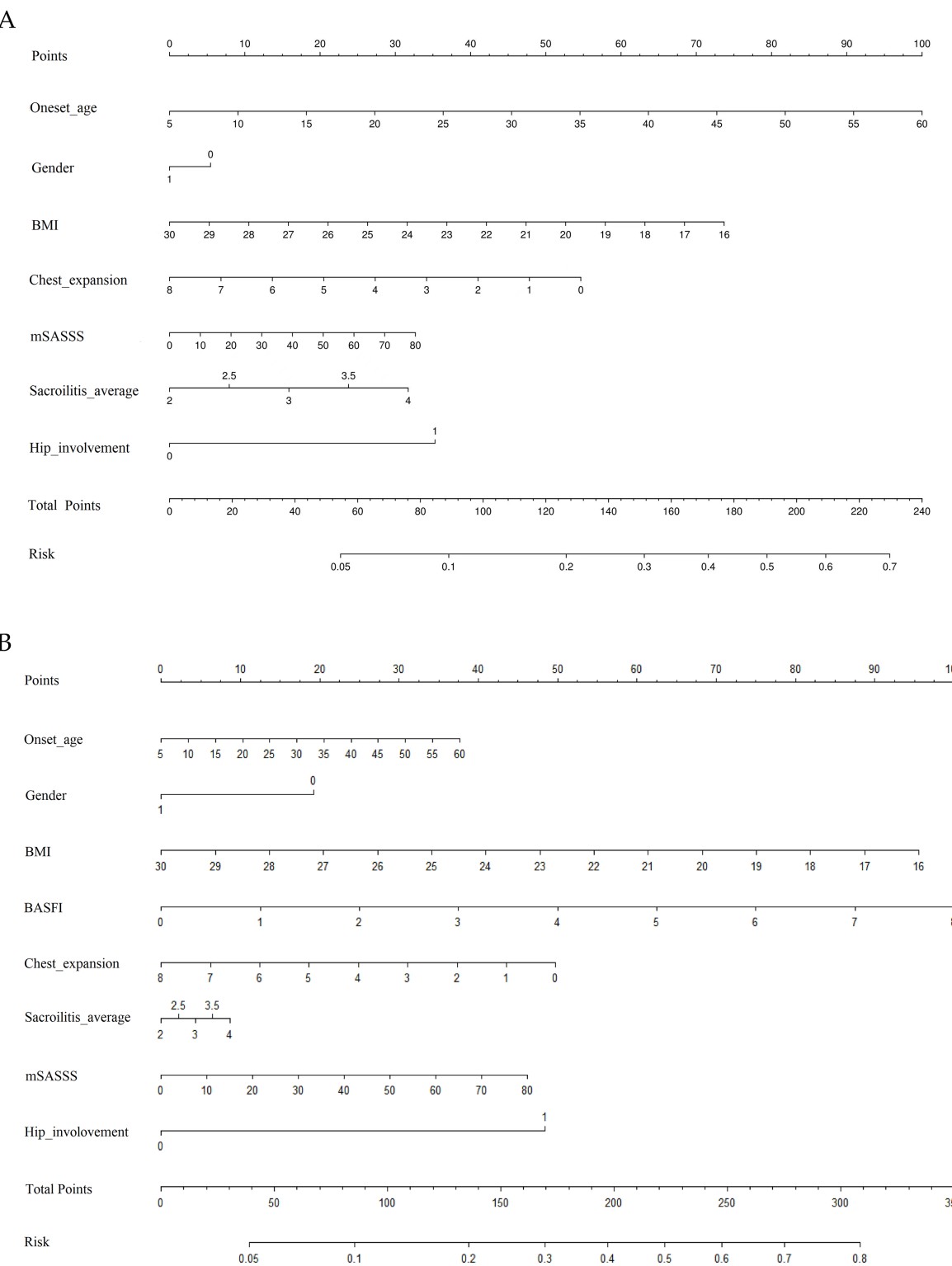

**Figure 2  Risk prediction nomograms for low BMD in AS patients at two different sites.** (A) Femoral neck; (B) total hip. BMI, body boss index; BASFI, Bath Ankylosing Spondylitis Functional Index; mSASSS, modified ankylosing spondylitis score; Sacroiliitis average, means average radiological grade of the sacroiliac joint.

low BMD in different sites based on an interaction analysis and nomogram prediction model. Our interaction analyses revealed that low BMD could be caused by the superposition of risk factors. Our interaction analyses revealed that multiple factors that disrupt bone metabolism balance increase the risk of low BMD in an additive manner. Identifying these risk factors for low BMD in AS patients will prompt early intervention to prevent fractures.

Currently, it is still a controversial topic about the common sites of low BMD in AS patients (*Klingberg et al., 2012*; *Singh et al., 1995*; *Wang et al., 2017*; *Cai et al., 2020*; *Deminger et al., 2017*). *Deminger et al. (2017)* found that low BMD was more common at the proximal femur compared to the lumbar spine (16.5% *vs.* 6.3%) in AS patients. However, *Klingberg et al. (2012)* suggested that the lumbar spine is the most common sites of low bone mass in AS patients. Unfortunately, these studies did not discuss the differences in the sites prone to low BMD between patients with and without hip involvement. Our study found that the total hip was the most common site of low BMD (34.5%) in patients with hip involvement, while the lumbar spine is the most common site of low BMD (16.7%) in patients without hip involvement. This suggested that inflammation of the hip and the resulting limitation of activity may accelerate the loss of hip BMD. Furthermore, the impact of osteophytes on bone density is also noteworthy, especially concerning lumbar spine BMD. *Kaya et al. (2009)* in a follow-up of 55 AS patients after 1 year, found that 3.4% of patients had an increase in lumbar spine BMD. It is believed that osteophytes have an effect on the measurement of anterior and posterior lumbar spine bone density, which was further corroborated by our study. For AS patients with osteophytes, the increase in mSASSS is significantly correlated with higher anterior and posterior lumbar spine BMD. Therefore, clinicians should take note that for patients with osteophytes, it is unreasonable to assess the extent of bone loss based solely on BMD measurements in the anterior and posterior lumbar spine positions.

Since predictors may have different effects on BMD at different sites, we explored potential risk factors for low BMD at the femoral neck and total hip separately. To avoid the potential effect of syndesmophytes on the measurement of BMD at the anteroposterior lumbar spine, we did not explore risk factors for low BMD at the lumbar spine. Our results revealed that chest expansion, mSASSS, BMI, the average radiographic grade of the sacroiliac joint, and hip involvement were the common risk factors of the femoral neck and total hip. These findings also confirm some previous studies. For example, the relationship between mSASSS and low BMD has also been explored in different literatures. *Karberg et al. (2005)* demonstrated that mSASSS score was significantly associated with low BMD, especially in the femoral neck. Another study also showed that low BMD was significantly associated with the development of new syndesmophytes (*Kim et al., 2018*). Based on the above results, it could be confirmed that AS patients with high mSASSS scores caused limited activity, which might accelerate the process of low BMD.

In addition, we found that more severe sacroiliitis was also a risk factor for low BMD in the femoral neck and total hip. In previous studies, it was also documented that low trabecular bone score in AS patients was associated with the severity of sacroiliitis (*Kang et al., 2018*). This may be related to trabecular bone loss as a result of chronic
inflammation, and its impact on BMD is manifested in a non-single site. Therefore, aggressive interventions in the progressive stages of AS (especially for more severe sacroiliitis) should effectively prevent low BMD by increasing the mobility associated with pain relief and potentially having a direct anti-inflammatory effect on bone (*Wang et al., 2017*). Furthermore, we found that hip involvement was a common risk factor for low BMD in the femoral neck and total hip. This result was supported by Wang et al, who found that hip involvement was one of the risk factors for developing bone loss in AS patients (*Wang et al., 2015*; *Liu et al., 2021*). On the one hand, the relationship between hip involvement and low BMD could be explained by local inflammation in the hip joint. On the other hand, for AS patients with hip involvement, early pain and late hip ankylosis would decrease the patient's activity, which would further aggravate bone loss.

Although the impact of each risk factor is small when the risk factors are evaluated separately, the impact is more obvious when there is a superposition of other risk factors. However, most of the current studies tend to focus on individual risk factors, while ignoring the interaction between other factors. To extend the outcomes reported by published studies, we explored the interaction between each risk factor and observed synergistic effects. Our results showed that the presence of syndesmophytes significantly increased the risk of low BMD in the femoral neck and total hip when the radiological average grade of the sacroiliac joint exceeded grade 3 or hip involvement was present. Notably, hip involvement not only interacted with syndesmophytes but also with the radiographic grade of the sacroiliac joint, highlighting the importance of severe sacroiliitis, syndesmophytes, and hip involvement in the development of low BMD. These results underscore the need to closely monitor AS patients with severe radiological damage, hip involvement, and higher sacroiliitis grades to prevent the occurrence of low BMD.

Taken together, through the analysis of the survey data, we explored the risk factors of low BMD in different sites in AS patients. More importantly, we found that some risk factors may act on the susceptibility of low BMD in a cumulative way. Finally, to facilitate the practical application of our findings, we created a risk prediction nomogram model for the occurrence of low BMD at different sites for AS patients, which revealed reasonable prediction accuracy. We hope that this study provides background data that can be used to further explore the potential risk factors of low BMD in AS patients, including individual and interaction effects, and further explore the risk factors of low BMD in AS patients for the process of early detection and prevention.

There are some limitations in this study. Due to our limited sample size, the conclusions are not definitive and therefore it is necessary to formulate corresponding standards according to different ages and gender. It is also necessary, for the future, to adopt quantitative computed tomography to avoid the interference of actual density in AS patients with syndesmophytes to draw more accurate conclusions (*Deminger et al., 2022*). Currently, the use of this model relies on certain radiological examination and measurement indicators. Despite these parameters being part of routine examinations, there is still room for further enhancement in the convenience of applying this model. Furthermore, there is literature mentioning the beneficial effects of TNF-α inhibitors on BMD (*Haroon et al., 2014*). Unfortunately, in this study, there were only eight cases of

patients treated with TNF-α therapy in the training set, which resulted in the inability to observe the impact of TNF inhibitors on BMD. The major strength of the present study is the finding that AS patients with hip involvement are more likely to experience low BMD in the total hip, whereas those without hip involvement are more prone to low BMD in the lumbar spine. We also identified several risk factors associated with low BMD in the femoral neck and total hips, including syndesmophytes, hip involvement, and radiological average grade of the sacroiliac joint. Importantly, we found that these factors increase the risk of low BMD in an additive manner. Finally, we established an effective prediction model in order to facilitate new research into the possibility of creating a prevention strategy.

## CONCLUSION

Low BMD was most likely to occur in the total hip in patients with hip involvement and in the lumbar spine in patients without hip involvement. This study identified syndesmophytes, hip involvement and severe sacroiliitis increases the risk of low BMD in an additive manner and established a nomogram prediction model to help rheumatologists identify high risk patients to prevent low BMD.

## ACKNOWLEDGEMENTS

We thank the patients from the Department of TCM rheumatology, China-Japan Friendship Hospital for providing support to this study, the participants of cedars Sinai Medical Center for their guidance and assistance with this study, and all the investigators who contributed to this study.

### Funding

The study received funding from the National High Level Hospital Clinical Research Funding (No. 2022-NHLHCRF-LX-02-0104), the National Science Foundation of Beijing, China (No. 7182148) the Beijing Traditional Chinese Medicine Science and Technology Development Fund Project (No. JJ-2020-87) and the Spondylitis Association of America. The funders had no role in study design, data collection and analysis, decision to publish, or preparation of the manuscript.

### Grant Disclosures

The following grant information was disclosed by the authors:
National High Level Hospital Clinical Research Funding: 2022-NHLHCRF-LX-02-0104.
National Science Foundation of Beijing, China: 7182148.
Beijing Traditional Chinese Medicine Science and Technology Development Fund: JJ-2020-87.
Spondylitis Association of America.

### Competing Interests

The authors declare that they have no competing interests.

## Author Contributions

- Wenting Sun conceived and designed the experiments, performed the experiments, analyzed the data, prepared figures and/or tables, authored or reviewed drafts of the article, and approved the final draft.
- Wenjun Mu conceived and designed the experiments, performed the experiments, analyzed the data, authored or reviewed drafts of the article, and approved the final draft.
- Caroline Jefferies performed the experiments, prepared figures and/or tables, and approved the final draft.
- Thomas Learch analyzed the data, prepared figures and/or tables, and approved the final draft.
- Mariko Ishimori analyzed the data, prepared figures and/or tables, and approved the final draft.
- Juan Wu performed the experiments, analyzed the data, authored or reviewed drafts of the article, and approved the final draft.
- Zeran Yan performed the experiments, analyzed the data, authored or reviewed drafts of the article, and approved the final draft.
- Nan Zhang performed the experiments, analyzed the data, authored or reviewed drafts of the article, and approved the final draft.
- Qingwen Tao performed the experiments, authored or reviewed drafts of the article, and approved the final draft.
- Weiping Kong conceived and designed the experiments, prepared figures and/or tables, authored or reviewed drafts of the article, and approved the final draft.
- Xiaoping Yan conceived and designed the experiments, authored or reviewed drafts of the article, and approved the final draft.
- Michael H. Weisman conceived and designed the experiments, authored or reviewed drafts of the article, and approved the final draft.

## Human Ethics

The following information was supplied relating to ethical approvals (*i.e.*, approving body and any reference numbers):

The research ethics committee of China-Japan Friendship Hospital (approval No. 2017-67) and the ethics committee of Cedar Sinai Medical Center (approval No. pro00048849).

## Data Availability

The raw measurements are available in the Supplemental File.

## Supplemental Information

Supplemental information for this article can be found online at http://dx.doi.org/10.7717/peerj.16448#supplemental-information.

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
