# Peer review of "Interaction effects of significant risk factors on low bone mineral density in ankylosing spondylitis"

_PeerJ, doi:10.7717/peerj.16448_

## Round 0.1 · original submission · Major Revisions

A number of issues have been identified that need to be addressed, particularly in statistical analyses.

·

Basic reporting

Basic Reporting:
• The article is written clearly in professional English.
• The introduction and background section is inadequate, particularly with reference to existing literature in this area. For instance, the literature review by Van der Wiejden (2012) is briefly cited by the authors, but they have not emphasised their important conclusion that BMD loss occurs in early AS. The literature review at this point is a bit deficient. They have not cited relevant literature reviews of bone mass in axSpA by Kilic (2015) and Ramirez (2018). The authors have not cited a relevant review by Haroon (2014) that suggested a beneficial effect of anti-TNF therapy on BMD in AS. Again, this should also be referred to in the discussion, as their study had a low proportion of subjects on biologic therapy. With respect to more focused articles relating to predictive factors for low BMD in AS, the authors have appropriately cited a study by Kaya (2009) which concluded that with progression of AS, BMD at the hip decreases whilst BMD at the spine increases – but this study should be referred to again in the discussion, as their findings confirmed this association. The study by Kang (2018) was appropriately referred to in the discussion. but not the introduction. They had reported that the grade of sacroiliitis, mSASSS, syndesmophytes were strongly linked to low BMD – also (marginally) gender, BMI, and ESR. The authors have also cited a large controlled study of BMD in Chinese AS patients by Wang (2015) which found that BMD was related to hip involvement and high disease activity (represented by ESR and CRP).
• The structure conforms to PeerJ standards. The clarity of the methods and discussion sections could be improved.
• Two figures are provided. I don’t understand Figure 1 and I don’t think clinicians will be able to make sense of this. A multiple regression analysis would perhaps be more useful approach. Figure 2 presents a nomogram to estimate the risk of low BMD – this might be useful in clinical practice but it requires a lot more explanation as to how it was derived and how it should be used. I assume that a vertical line should be drawn from each of the risk factor score scales up to the ‘points’ line and then the total points are added up and linked to the risk line in some way. BMI is a general risk factor for BMD, but for some reason age is presented as age of onset rather than current age. Gender appears to have very little effect on BMD, which is not what is known from large studies of the normal population.
• The raw data has been supplied. I checked the excel file for their calculation for duration of disease and the data matched their calculation (the onset age was defined using the onset of symptoms not diagnosis). Osteoporotic BMDs were uncommon, some BMDs were very high. Data on vertebral fracture incidence appears to be missing.

Experimental design

• This research study appears to have been designed to ascertain the main factors determining BMD in AS patients and to develop a predictive scoring tool to identify patients with low BMD. Although other studies have identified similar risk factors, the attempt to develop a clinical nomogram is novel and potentially of use to clinicians. This aim is not clearly articulated in the introduction (line 71): surely the nomogram would be used to guide clinicians dealing with AS patients as to which of their patients need a DXA scan/treatment for osteoporosis. However, a number of radiographs would need to be carried out to complete the nomogram, so it may not be of practical use (this could be addressed in the discussion).
• The study received ethical approval, and I have no concerns about ethical aspects of this study.

Validity of the findings

The pragmatic value of this study is limited. Even though the authors have chosen to use the NY classification criteria for AS, the participants were young with a short duration of disease few were on anti-TNF drugs. Even if the statistical validity of this approach were to be verified, the nomogram they propose would be complex for clinicians to implement and would potentially expose patients to more radiation than performing a DXA scan.

Additional comments

The authors are to be congratulated for carrying out a detailed study of the factors that influence bone mineral density in AS patients. The general conclusions will be informative for many clinicians. However, a diagnostic tool does have to be carefully validated, and I'm not convinced that in its current form the nomogram is ready for clinical use.

Reviewer 2 ·

Basic reporting

Some English language editing is needed, especially in the section on statistical analysis.

Experimental design

Data Collection/Measurements
Kindly cite the references for BASDAI and BASFI. Kindly cite the original references for ASDAS and ASDAS CRP.

Statistical Analysis
“Categorical and continuous variables are expressed as median, range and/or mean, or standard deviation (SD), where appropriate.”
Comment: Categorical variables are expressed as frequencies or percentages, which has not been mentioned.

Validity of the findings

Results:

It has been mentioned in the methods that the low BMD group was also divided into osteoporosis and osteopenia. Please mention how many patients had osteopenia or osteoporosis.

According to table 1,
Age at disease onset and gender distribution were not significantly different between AS patients with and without low bone mineral density. Please explain why age at onset and gender were included in the risk prediction nomograms for low BMD in AS (figure 2).

Figure 1:
Some readers may not be familiar with the decision curves. In the context of diagnosis of low bone mineral density, please explain what is meant by net benefit, treat all and treat none.

In figure 2, please define what gender 0 or 1 signify.

Reviewer 3 ·

Basic reporting

The objective of the study were clearly stated. The introduction section need more information. I would expect a little detailing of techniques used to quantify the data. And how the current study different from other studies.

Experimental design

The author should add more detail about the trial design, such as when the bone mineral density were collected. Also, as author mention in introduction, "the occurrence of low BMD is quite hidden" the author may want to add discussion about the data issues and the potential missing data issue in the current study. Can the authors comment on the potential selection bias.

Validity of the findings

The "Identification of Risk Factors for Low BMD in the Femoral Neck and Total Hip" need to add more information. The author should add more discussion about the multiple testing issue when hypothesis testing in the univariate logistic regression analyses were done, there are multiple literature could be added for FDR control. Also, with the adjusted model, the author may want to report AIC and BIC for the model to avoid overfitting.

The interaction explorations were done based on the p-value, where as mentioned above multiple testing issue should be considered. The author could use BH method to adjust the p-values.

The prediction model section need more information. The author should add model validation and prediction accuracy.

Additional comments

Overall, potential missing data and selection bias problem may be discussed.

Reviewer 4 ·

Basic reporting

- The authors provided a clear and comprehensive description of the study design, outlined the recruitment process and inclusion criteria for participants, and details of the collection process of each variable are provided.

- In table S3, AUROC value for each model is not provided.

- Line 248, replace ‘studys’ with ‘studies’

- In Table 2, replace ‘lumber spine’ with ‘lumbar spine’

Experimental design

- In the statistical analysis section, more details need to be added to explain the process of building and validating the nomogram model. It is important to provide a clear and comprehensive description of the steps taken to build the nomogram model. Consider including the details of the statistical techniques used, variable selection procedures, and any assumptions made during the process. Also, a thorough explanation of the validation strategy used to assess the performance of the nomogram model. Clearly outline the metrics employed, such as accuracy, sensitivity, specificity, or area under the curve (AUC). Include information on the choice of training and test datasets, cross-validation techniques, and any external validation, if applicable.

Validity of the findings

- In the prediction accuracy assessment, the full logistic model was compared with the basic model which includes all variables except for the significant risk factors identified by regression analyses. It has been shown in Figure S3 that the full model is superior to the basic model, however, how to ensure the full model is not overfitted? More details and discussion can be added.

- When identifying risk factors for low bone mineral density (BMD), there are multiple variables that may exhibit high correlation, such as BASMI, BASFI, and mSASSS, which are all associated with patients diagnosed with AS. The presence of such correlations among these variables raises concerns regarding the dependability of the reported coefficients and their interpretation. In the presence of strong correlations, it becomes challenging to determine the unique contribution of each variable in explaining the outcome. The coefficients may provide misleading information, suggesting that certain variables have a significant impact when, in reality, they might be capturing the effects of other correlated variables. To address the issue of collinearity, certain mitigation strategies can be employed.

---

## Round 0.2 · Minor Revisions

The authors have addressed the reviewers' comments. Some additional comments from the reviewer were raised.

·

Basic reporting

The article has been extensively revised and a serious attempt has been made to address all of the comments and suggestions by the reviewers.

There are still a few minor areas in which there is a lack of clarity in the text & I have suggested rephrasing where necessary.

The literature review is now acceptable and the methodology/results sections have been greatly improved. The raw data has been shared.

Minor revisions suggested:
1. Line 35 suggest replacing 'experience' with 'have'
2. Line 45 suggest ‘…identify AS patients who are at a high risk of developing osteoporosis at the hip’
3, Line 56 suggest omitting this line and including Kilic with the other references on BMD in AS
4. Line 57 suggest 'many physicians are unaware of the increased risk of osteoporosis in AS'
5. Line 209 Consider rephrasing or omitting this sentence. You’ve already indicated that the prediction accuracy of the full model was superior to the basic model.
6. Line 249 Consider revising and omitting the phrase 'due to multiple factors that disrupt bone metabolic balance' - low BMD contributes to increased fracture risk in AS: identifying risk factors for low BMD will prompt early intervention to prevent fractures.

Experimental design

The design is now clearly stated in the introduction & well documented in the methods.

Validity of the findings

Whilst this study is unlikely to change clinical practice, it is a more detailed analysis of BMD in AS patients than has been attempted in the past and their conclusions are supported by the evidence they present.

Reviewer 3 ·

Basic reporting

The author add relative information and more analyses to resolve my previous comments. No further comments.

Experimental design

No further comments

Validity of the findings

No further comments

Reviewer 4 ·

Basic reporting

I think that the authors have adequately addressed the comments made by the reviewers in the revised version of the manuscript. Therefore, I have no further comments.

Experimental design

I think that the authors have adequately addressed the comments made by the reviewers in the revised version of the manuscript. Therefore, I have no further comments.

Validity of the findings

I think that the authors have adequately addressed the comments made by the reviewers in the revised version of the manuscript. Therefore, I have no further comments.

---

## Round 0.3 · accepted · Accept

The authors have addressed the reviewers' concerns.